# The Versatile Attributes of MGMT: Its Repair Mechanism, Crosstalk with Other DNA Repair Pathways, and Its Role in Cancer

**DOI:** 10.3390/cancers16020331

**Published:** 2024-01-11

**Authors:** Qingming Fang

**Affiliations:** Department of Biochemistry and Structural Biology, Greehey Children’s Cancer Research Institute, University of Texas Health San Antonio, San Antonio, TX 78229, USA; fangq1@uthscsa.edu

**Keywords:** MGMT, AGT, base excision repair (BER), mismatch repair (MMR), direct reversal, homologous recombination (HR)

## Abstract

**Simple Summary:**

This review offers a thorough examination of the potential role and structural properties of MGMT, the progression in targeting MGMT, and the interactions between MGMT and DNA repair pathways involved in processing DNA lesions.

**Abstract:**

O^6^-methylguanine-DNA methyltransferase (MGMT or AGT) is a DNA repair protein with the capability to remove alkyl groups from O^6^-AlkylG adducts. Moreover, MGMT plays a crucial role in repairing DNA damage induced by methylating agents like temozolomide and chloroethylating agents such as carmustine, and thereby contributes to chemotherapeutic resistance when these agents are used. This review delves into the structural roles and repair mechanisms of MGMT, with emphasis on the potential structural and functional roles of the N-terminal domain of MGMT. It also explores the development of cancer therapeutic strategies that target MGMT. Finally, it discusses the intriguing crosstalk between MGMT and other DNA repair pathways.

## 1. Introduction

Chemical insults, both endogenous (e.g., S-adenosylmethionine) and exogenous (e.g., alkylating agents), constantly attack cellular DNA, resulting in DNA lesions [1,2,3,4]. Among these chemicals, alkylating agents play a significant role in causing DNA damage [5]. These agents are widely present in the environment, used as anticancer compounds in clinical settings [1,2,3,4,6,7,8,9,10], and can be endogenously generated during normal cellular processes. For example, S-adenosylmethionine, a methyl donor involved in various cellular reactions, has been shown to induce DNA methylation damage [3,11,12]. The attack of alkylating agents on DNA can lead to different types of lesions on the heterocyclic bases or backbone [3,5,13,14,15,16]. Most of these resulting adducts are mutagenic or toxic, prompting cells to develop various proteins for their detection and repair [13,17]. Interestingly, many of these alkylation lesions are repaired through direct removal of the adduct [5]. O^6^-alkylguanine-DNA alkyltransferase or O^6^-methylguanine-DNA methyltransferase, known as AGT or MGMT, is an enzyme responsible for the repair of alkylated DNA through a process known as “suicide” repair [3,4,13]. MGMT expression is primarily regulated through epigenetic modifications [18,19,20], and extensive research has indicated that reduced MGMT expression is attributed to the methylation of the CpG island that spans 762 base pairs and contains 98 CpG sites found in the MGMT promoter region [21,22,23,24]. Mechanistically, the primary function of MGMT involves the transfer of the methyl group located at the O^6^ site of guanine to cysteine residues located within the MGMT active site, thereby preventing mutations, cell death, and the development of tumors caused by alkylating agents [3,13]. Apart from being an avenue for cancer therapeutic strategies, MGMT has also been investigated as a valuable research tool for the specific labeling of proteins [5,13,25,26,27,28,29,30].

This review covers three main areas of focus. First, it explores the structural role of the N-terminal domain of MGMT. Second, it delves into the development of cancer therapeutic strategies that target MGMT. Lastly, it examines the crosstalk between MGMT and other DNA repair pathways. Readers are directed to several excellent reviews and articles that have been published [3,4,5,13,14,15,24,31,32,33,34,35,36,37,38,39].

## 2. Repair Mechanism

The initial fully characterized MGMT or AGT was the Ada gene product from *E. coli* [40,41]. It performs the repair of O^6^-methylguanine (m^6^G) in DNA through the direct transfer of the methyl group from the DNA to a cysteine residue within the protein itself (Figure 1) [40,41]. The inducibility of the Ada protein in response to alkylation damage facilitated its biochemical examination, enabling purification in substantial quantities after such treatment [42,43]. Subsequent investigations unveiled another MGMT in *E. coli*, Ogt [44,45]. Unlike the Ada protein, Ogt is not inducible by alkylating agents and has more limited expression levels. Both proteins are commonly referred to MGMT.

In the early 1990s, accumulating evidence indicated the existence of a similar protein in mammalian tissues, leading to the cloning of human MGMT (hMGMT) [46,47]. Studies have demonstrated that human MGMT is highly effective in repairing larger and bulky adducts, such as O^6^-[4-oxo-4-(3-pyridyl)butyl]guanine, which is formed by the tobacco-derived carcinogen 4-(methylnitrosamino)-1-(3-pyridyl)-1-butanone (NNK) [48], as well as seven- or four-carbon O^6^-2′-deoxyguanosine-alkyl-O^6^-2′-deoxyguanosine interstrand DNA cross-links [49,50]. However, it exhibits low efficiency in repairing O^4^-methylthymine [51,52]. For more information on MGMT substrates, please refer to reference [13].

MGMT is widely present in almost all organisms. It is easily identifiable due to the characteristic protein sequence surrounding the central Cys site that acts as the acceptor for alkyl groups: (I/V)PCHR(V/I)-(I/V) (Figure 1) [13]. Multiple structures of hMGMT have been elucidated, providing valuable insights into its functioning [53,54,55,56]. The structure of MGMT comprises two domains. The C-terminal domain encompasses the active site pocket and DNA binding region, interconnected by a hinge formed by the conserved residue Asn137 (amino acid numbers provided refer to hMGMT) (Figure 1) [53,54,55,56]. The N-terminal domain plays a vital role in maintaining structural stability, and can stabilize the complex formed when the two domains are expressed separately [57]. The initial published structure revealed a similar fold to the C-terminal domain of Ada (C-Ada: One of MGMTs in *E. coli*), even in the N-terminal domain, despite a lower sequence similarity between the two proteins [53](Figure 1). Another native structure, along with the benzylated form, uncovered an intriguing revelation—the N-terminus of hMGMT (N-hMGMT) contains a zinc(II) ion coordinated by Cys5, Cys24, His29, and His85, arranged tetrahedrally [54] (Figure 2). These four residues are conserved across mammalian sequences of the protein but are absent in yeast (*S. cerevisiae*) and bacterial homologues [58].

Further biochemical investigations have revealed that although the zinc(II) site in MGMT is located away from the reaction center, its binding to a zinc(II) ion effectively lowers the pKa of Cys145 compared to the apo protein, thereby enhancing the reactivity of the protein [59]. hMGMT utilizes a helix-turn-helix motif to interact with DNA, but exhibits an unusual behavior of specifically targeting the minor groove [55]. The recognition helix, characterized by hydrophobic residues, tightly fits into the minor groove with minimal sequence-specificity [55]. The structural findings have led to the proposal of a comprehensive mechanism for the activation and repair of Cys145. This mechanism involves a hydrogen bonding network that activates the Cys145 residue, with a crucial Tyr residue (Tyr114) playing a role in identifying the damaged guanine, and providing a proton to facilitate the removal of the alkyl adduct. Additionally, an arginine finger (Arg128) stabilizes the displacement of the base in the DNA helix (Figure 2) [55,60]. These residues are conserved across different MGMTs. The preference of hMGMT for larger aromatic substrates is evident in the benzylated structure, where the benzyl group, covalently attached to the active site cysteine, is positioned between Pro140, Ser159, and Tyr158 through hydrophobic interactions (Figure 2) [54]. In contrast, in C-Ada, the proline residue is replaced by alanine, and a glycine residue (Gly160) on the opposite side of the hMGMT pocket is substituted with tryptophan. This structural difference in C-Ada results in a narrower substrate binding pocket (Figure 1), which explains its inclination towards smaller lesions [58].

## 3. The Function of N-Ada and the Potential Role of N-hMGMT

The Ada protein consists of two domains: a 20 kDa N-terminal domain (N-Ada) and a 19 kDa C-terminal domain (C-Ada) containing 354 amino acids (Figure 1) [61,62]. In *E. coli*, the Ada protein serves multiple functions. N-Ada is responsible for repairing the Sp-configurated methylphosphotriester, while C-Ada repairs the highly mutagenic m^6^G and m^4^T [61,62,63]. Additionally, Ada acts as a transcriptional activator that induces the robust expression of the ada, alkA, alkB, and aidB genes, thereby regulating the adaptive response to methylation resistance [64]. This response occurs after transferring the methyl group from the Sp-methylphosphotriester to the Cys38 residue [63]. The methylated N-Ada transforms into a potent DNA binder, enabling it to recognize the promoter regions of the Ada regulon and recruit RNA polymerase to initiate the transcription of the aforementioned four genes [17,63,65,66,67]. This specific Cys38 residue demonstrates selective activation, for it is the only Cys ligand that is not involved in any hydrogen-bonding interactions, and exhibits higher nucleophilicity compared to other Cys residues (Figure 3) [63,68,69,70]. Additionally, this residue acts as a ligand to a zinc(II) ion present in the active site, which is also bound by three other Cys residues (Figure 3) [63,68,69,70]. A Zn cluster with four Cys acquires negative charges. Within this cluster, a Cys residue, potentially in a transiently dissociated state from the zinc(II) center, but remaining deprotonated, is capable of attacking a methylphosphotriester (Figure 3). This attack results in the transfer of the methyl group [63]. This zinc ion also plays an essential role to keep the structure of N-Ada [63].

Currently, no N-Ada homologues have been identified in humans, and there is no evidence to suggest that methylphosphotriesters are repaired in human cells. The adaptive response to the methylation challenge is not clearly observed in eukaryotes. All the residues identified as being associated with the activity and DNA binding of hMGMT are located in the C-terminal domain (Figure 2) [54,55,58]. Structural and biochemical studies of hMGMT indicate that N-hMGMT may primarily serve a passive structural role, rather than directly participating in the protein’s functional activity [53,54,55,56]. Previous biochemical studies indicated that hMGMT exhibits more similarity to Ogt than to Ada [71,72]. However, the published structures of hMGMT have revealed a similar fold to Ada [53,54,55,56] (Figure 1 and Figure 4). The role of N-hMGMT remains unclear. When the two domains are expressed separately, we found that N-hMGMT plays an important structural function in stabilizing the C-terminal domain of hMGMT (hMGMT-C) and forming the N-hMGMT/hMGMT-C complex [57]. Unexpectedly, we discovered that N-hMGMT exhibits Zn^2+^-dependent DNA repair activity in vitro, and can repair m^6^G DNA damage but not methylphosphotriester damage [57]. N-hMGMT shows very weak activity toward double-stranded DNA (dsDNA) containing m^6^G, while it is highly sensitive to MGMT inhibitor-O^6^-BenzylGuanine (O^6^-BG) [57], suggesting that N-hMGMT may prefer m^6^G in single-stranded DNA (ssDNA) or the dGTP pool rather than dsDNA. This conclusion is consistent with the crystal structures of hMGMT, which show that all the residues related to DNA binding are located in the hMGMT-C. Although the structural difference in C-Ada results in a smaller substrate binding pocket, this may explain why hMGMT is more sensitive to large DNA lesions such as O^6^-BG [58]. However, the active site of hMGMT-C is not located on the protein surface. How can O^6^-BG be recognized and located in the active site of hMGMT-C? Evidence suggests that the binding of zinc in N-hMGMT plays an essential role in recognizing O^6^-BG. N-hMGMT is more sensitive to O^6^-BG than hMGMT, and the loss of zinc results in the inactivation of N-hMGMT and the N-hMGMT/hMGMT-C complex (see Table 1 and Figure 2).

To explore the potential role of N-hMGMT, we employed protein structure prediction online server (Phyre2, Imperial College, London, UK). The results showed that the highest scoring template, with 99.6% confidence, indicates that the structure of N-hMGMT exhibits a ribonuclease H-like motif (Table 2). Genotoxic agents have the ability to cause damage to dsDNA, ssDNA, RNA, and the dGTP pool, leading to the formation of DNA and RNA adducts [14,74,75,76,77,78]. Interestingly, it has been observed that higher levels of methylation occur in RNA compared to DNA in human cells treated with N-methyl-N-nitrosourea (MNU) [79]. Mammalian cells have developed repair mechanisms to address alkylation damage in RNA [80,81]. The existence of RNA repair mechanisms serves as an important defense mechanism for cells, helping them mitigate dysfunctional RNA resulting from alkylated damage [76,77,78,82]. In hMGMT, the repair of RNA is impeded by a steric clash between the 2′-hydroxyl group and the Cα atom of Gly131. This clash hinders the ability of hMGMT-C to efficiently repair RNA lesions [5,56]. The structural characteristics of N-hMGMT suggest that it may have a propensity for repairing RNA lesions despite of the inefficiency of hMGMT-C, but additional research is required to establish its actual repair capabilities and preferences. It is unknown whether the alkylated N-hMGMT plays a similar role as alkylated N-Ada.

## 4. The Fate of Alkylated MGMT

Early reports suggested that the addition of an alkyl group to the Cys145 of hMGMT causes structural alterations, leading to its recognition by ubiquitin ligases and subsequent degradation by the proteasome [83,84,85]. This degradation process may be necessary for continuous repair, as the presence of alkylated MGMT or inactive mutants like C145A can interfere with the repair function of active MGMT molecules [86]. It has been proposed that the disruption of a salt bridge between Asn137 and the carbonyl oxygen of Met134, caused by a steric clash with the newly formed S-alkylcysteine, serves as the signal for this degradation [54]. These alterations destabilize the protein but have minimal impact on DNA binding [87,88].

Accumulating evidence suggests that alkylated MGMT functions as a transcriptional regulator, regulating the DNA damage response. hMGMT is found at active transcription sites, facilitated by interacting with RNA polymerase II, and it exhibits a preference for repairing m^6^G damage specifically in the transcription strands [89]. Upon alkylation, MGMT undergoes a conformational change that exposes VLWKLLKVV residues (codons 98 to 106), allowing it to bind to the estrogen receptor (ER), a critical transcription factor involved in regulating cell proliferation [90]. The binding of alkylated MGMT to ER prevents its interaction with steroid receptor coactivator-1, thereby inhibiting the activation of ER-regulated gene expression [90]. This suggests that alkylated MGMT functions as a transcription regulator, leading to cell cycle arrest. Additionally, alkylated MGMT suppresses the expression of deubiquitinating enzyme 3 (DUB3), thereby impacting chemoresistance in ovarian cancer [91]. It is currently unknown whether this effect is associated with alkylated N-hMGMT or alkylated hMGMT-C. Further research is necessary to elucidate this relationship. Previous studies have indicated that the expression of hMGMT can be modulated by glucocorticoids and partial hepatectomy [92,93]. The promoter region of hMGMT has been disclosed to contain estrogen-responsive elements and antioxidant-responsive elements recently, and the expression of hMGMT can be influenced by estrogen and nuclear factor-erythroid 2-related factor-2 (Nrf-2) [94]. The adaptive response triggered by alkylating agents is not clearly observed in mammalian cells [13]. However, clinical data has shown that chemotherapy utilizing alkylating agents may lead to an up-regulation of hMGMT expression in gliomas, although the underlying mechanisms remain unclear [24,95,96]. The factors that determine the different fates of alkylated MGMT remain unclear. One potential clue is that alkylated hMGMT has a half-life of several hours, suggesting the involvement of regulation mechanisms within this time period [13]. Further research is needed to uncover the conformational changes of alkylated hMGMT, its interactions with other proteins, and the detailed mechanism of ubiquitination. Another clue pertains to the post-translational modification (PTM) of hMGMT. Protein kinases can phosphorylate hMGMT [97], and PARP1 can cause the PARylation of MGMT in response to alkylating agent treatment [98,99]. While these modifications have an impact on its function [98,99,100], additional research is required to determine whether these modifications affect the stability and conformational alterations of hMGMT.

## 5. The Role of hMGMT in Cancer Prevention and Chemotherapy

The induction of O^6^-alkylGuanine (O^6^-alkylG) by alkylating agents can result in significant biological consequences such as mutagenicity, cytotoxicity, and carcinogenesis if not repaired by MGMT [13]. The expression of hMGMT in various cultured cell lines has been shown to effectively reduce GC to AT mutations and cytotoxicity caused by alkylating agents [36,101,102,103,104,105]. Several excellent reviews have focused on the role of hMGMT in cancer prevention and chemotherapy [13,24,37,106,107]. We briefly highlight the important findings from animal models and clinical trials here. N-nitroso compounds, which are present in many foods, play a crucial role in the development of colorectal cancer [107]. The study utilized a transgenic mouse model where mice had high expression of MGMT in the colon [108]. These mice demonstrated reduced rates of aberrant crypt foci (ACF) formation following intraperitoneal administrations of the alkylating agent azoxymethane (AOM). Additionally, the overexpression of MGMT provided protection against G:C to A:T mutations in the KRAS oncogene induced by AOM [108]. Consistent with these findings, depleting MGMT in rats using the potent MGMT inhibitor B^6^G resulted in an increased frequency of colonic tumors following AOM treatment [109]. MGMT knockout mice, in an inflammation-driven colon carcinogenesis model induced by AOM and dextran sodium sulfate, exhibit high susceptibility to colon tumorigenesis [110,111]. NNK, a carcinogenic N-nitrosamine found in tobacco and tobacco smoke, leads to the development of lung tumors [112,113]. However, A/J mice with high levels of MGMT display smaller lung tumors and a lower frequency of K-Ras mutation, indicating reduced susceptibility to NNK-induced carcinogenesis [114,115]. The protective role of MGMT has also been observed in animal models of thymic lymphoma [116,117], hepatocellular carcinoma [118,119,120], skin cancer [121,122,123], and brain cancer [124,125,126].

DNA alkylating agents, such as methylating and chloroethylating agents, have been used in cancer therapy for over four decades [31,37,39]. Among these agents, the m^6^G and O^6^-chloroethylG products are the main toxic lesions in most cases, particularly at lower doses [39]. However, at higher doses of these agents, N-alkyl purines also contribute to cellular cytotoxicity, and the significance of MGMT in terms of overall cytotoxicity diminishes [39]. The MGMT gene plays a critical role in repairing DNA damage caused by alkylating agents, thereby promoting resistance to chemotherapy. The methylation of the MGMT promoter leads to the reduced or absent expression of hMGMT by inhibiting transcription, thereby increasing sensitivity to alkylating agents [127]. The methylated MGMT promoter has garnered substantial support as a predictive marker for the effectiveness of temozolomide (TMZ), an alkylating agent used in treating glioblastoma and low-grade gliomas. Multiple studies have provided evidence supporting this notion [128,129,130,131,132,133,134,135]. Additionally, two distinct clinical trials have demonstrated that the methylation status of the MGMT promoter can serve as an indicator of the prognosis of glioma patients treated with TMZ, but not when treated with the chemotherapy regimen comprising procarbacine, lomustine (BCNU), and vincristine (PCV) [136,137]. Moreover, clinical data has revealed a significant difference in the overall median survival among patients with malignant astrocytoma treated with BCNU, depending on their MGMT levels [138]. This finding highlights that MGMT can also function as a predictive marker for the treatment of malignant astrocytoma with chloroethylating agents. In clinical trials involving anaplastic oligodendroglioma patients, it has been observed that the methylation of the MGMT promoter is associated with improved overall survival and progression-free survival, regardless of whether patients underwent radiotherapy alone or sequential radiotherapy and chemotherapy with PCV [139]. The combination of MGMT promoter methylation and an isocitrate dehydrogenase 1 (IDH1) mutation was found to reduce the risk of progression in anaplastic glioma patients [140].

Emerging evidence indicates that hMGMT plays a role in the chemoresistance of cisplatin. In ovarian cancer, MCL1 is stabilized by MGMT-activated DUB3, leading to resistance to platinum/paclitaxel-based chemotherapy in ovarian cancer cells [91]. Conversely, in gastric cancer, a study has shown that hMGMT inhibits the autophagy-related gene (ATG) 4B, resulting in the suppression of autophagy. Cisplatin, on the other hand, counteracts MGMT-mediated autophagy suppression, thereby reducing chemosensitivity in gastric cancer. High MGMT expression and low ATG4B expression are significantly associated with survival in gastric cancer [141]. It is noteworthy that the impact of MGMT levels appears to differ in ovarian cancer and gastric cancer. However, further investigations are warranted to validate these findings.

## 6. The Development of Strategies That Target MGMT

hMGMT plays a crucial role in repairing DNA damage induced by methylating agents like TMZ, as well as chloroethylating agents such as BCNU, ACNU, and MeCCNU, thereby promoting resistance to chemotherapy using these agents [3,4,13,31,39]. hMGMT activity has also been associated with the resistance to 6-thioguanine that a medication used to treat leukemia [142]. 6-thioguanine is a poor substrate for hMGMT but binds to hMGMT after it is methylated to form S^6^-methylthioguanine [143,144,145,146]. Therefore, it is plausible that the binding of 6-thioguanine to hMGMT contributes to cellular resistance to this compound. Based on these findings, inhibiting MGMT shows potential in overcoming resistance to alkylating agents [3,4,13,31,39]. Numerous compounds that inhibit MGMT activity have been synthesized and utilized as adjuvants to enhance the cytotoxic effects of alkylating agents.

### 6.1. Non Cancer-Selective MGMT Inhibitors

O^6^-BG and O^6^-(4-bromothenyl)guanine (O^6^-4-BTG) are analogs of m^6^G that serve as irreversible pseudosubstrates of MGMT [147,148,149,150]. These compounds have been utilized as inhibitors of MGMT in clinical trials [13,39]. O^6^-BG has gained significant use in sensitizing glioma cells to the alkylating agent TMZ [13,39,151,152]. It possesses the ability to penetrate the blood–brain barrier and deactivate MGMT [24]. Clinical trials combining O^6^-BG with TMZ have shown promise in delaying brain tumor recurrence and increasing survival time [153,154,155,156]. However, it is important to note that this approach carries an elevated risk of side effects such as hydrocephalus, cerebrospinal fluid leak, and brain infection [157]. Additionally, the positive effects of this treatment have been observed in other types of tumors, including melanoma, colon cancer, and lymphoma [24,158]. O^6^-4-BTG serves as a highly potent MGMT inhibitor, surpassing the effectiveness of O^6^-BG [149,150]. Data from in vivo and in vitro studies across various tumor types have demonstrated that O^6^-4-BTG efficiently deactivates MGMT, resulting in a notable enhancement of tumor sensitivity to TMZ [159,160,161,162]. Phase II clinical trials have further supported these findings, demonstrating that the combination of O^6^-4-BTG and TMZ more effectively inhibits hMGMT compared to TMZ alone in patients with melanoma, prostate cancer, primary CNS cancer, and colorectal cancer [163,164,165,166]. However, it is worth noting that both O^6^-BG and O^6^-4-BTG have been associated with increased myelosuppression, without significantly improving the response rate to TMZ [24,39,155,156,157,167].

Due to the limited water solubility of O^6^-BG, it is necessary to develop more soluble derivatives to enhance its bioavailability. One approach is the meta-substitution of the aminomethyl group on the benzyl moiety of O^6^-BG. This modification enhances the water solubility of the compound and results in a more potent inhibitory activity against MGMT [168]. Several other 6-(benzyloxy)pyrimidine derivatives have been synthesized as potential inhibitors of MGMT [13,169]. In addition to the previously mentioned MGMT inhibitors, other types of inhibitors have been discovered. Acrolein and chloromethyltriazoles are highly reactive molecules with nucleophilic sites that can react with cysteine residues, effectively inhibiting MGMT [170]. Another inhibitor, 6-carboxyfluorescein, acts on MGMT in a non-covalent manner [171]. Lipoic acid, a natural compound containing a disulfide structure, has demonstrated potent MGMT inhibition and can enhance the cytotoxicity of TMZ in colorectal tumor cells that are resistant to TMZ [172]. Nitric oxide and disulfiram function by inactivating hMGMT through their interaction with the active Cys145 residue of the protein [173,174]. These studies have provided clear evidence of the correlation between the inhibitory effects of these compounds on MGMT and the increased cytotoxicity of chemotherapeutic drugs [37].

The primary concern regarding the use of non-cancer selective MGMT inhibitors is the increased risk of myelosuppression in bone marrow cells and other normal cells, which can lead to severe hematological toxicity such as leukemia and myelodysplastic syndrome [13,24,169,175]. To address these concerns and for various other reasons, researchers are actively developing cancer-selective inhibitors.

### 6.2. Cancer-Selective MGMT Inhibitors

The primary strategy for developing a cancer-specific MGMT inhibitor involves modifying the inhibitor with tumor-targeting groups [13,39]. The objective of this approach is to prevent MGMT inhibition in normal tissues while sensitizing cancer cells to chemotherapy [13,39]. Aerobic glycolysis is a prevalent metabolic characteristic observed in numerous tumors. In light of this, the conjugation of a glucose group to the MGMT inhibitor represents a promising concept for the development of cancer-selective MGMT inhibitors [176]. Studies have reported the high efficacy of both O^6^-BG-Glucose (O^6^-BG-Glu) and O^6^-BTG-Glu in inhibiting MGMT in various cancer cell lines, including T98G glioblastoma [150,176,177]. Furthermore, these agents have been shown to enhance the cytotoxic effect of temozolomide [150,176,177]. However, it should be noted that glucose conjugates are susceptible to transport out of the cell through ATP-binding cassette (ABC) transporter-mediated efflux, which can affect the efficiency of MGMT inhibition [177]. Folate receptors have shown great potential as carriers for the targeted delivery of chemotherapeutic drugs, particularly in squamous cell carcinomas, ovarian cancers, and certain non-small cell lung carcinomas, where they are frequently overexpressed [178,179]. The conjugation of a folate group to the MGMT inhibitor is used to develop specific MGMT inhibitors. O^4^-benzylfolate has demonstrated an MGMT inhibitory potency approximately 30 times higher than O^6^-BG. It has also been found to be effective in deactivating the P140K mutant MGMT, which is resistant to O^6^-BG-mediated inhibition [180]. Additionally, the efficacy of O^4^-benzylfolate in enhancing BCNU-induced cell death is dependent on the expression of an α-folate receptor [179]. In another approach, the synthetic 3′-γ-folate ester of O^6^-benzyl-2′-deoxyguanosine not only increased MGMT inhibitory activity in tumor cells but also sensitized HT29 and A549 cells to BCNU cytotoxicity [179]. Furthermore, this modification improved the water solubility of the inhibitor [179]. However, none of these cancer-selective MGMT inhibitors enters clinical trial.

Prodrugs have the potential to enhance tumor specificity and improve pharmacokinetic profiles [13,37]. Many solid tumors exhibit a characteristic of hypoxia, and hypoxia-activated O^6^-BG prodrugs have already been developed and utilized [181,182]. In particular, the release of β-glucuronidase is commonly observed from necrotic tumor cells. Therefore, designing O^6^-BG prodrugs that are substrate-related to β-glucuronidase could be a promising strategy to exploit [183].

### 6.3. Local Drug Delivery

Drug delivery plays a critical role in improving targeted therapy by utilizing diverse delivery systems and strategies to enhance the effectiveness and specificity of therapeutic agents [184]. Local drug delivery can achieve targeted therapy. Gliadel (BCNU wafers) marked the initial clinical application of polymer drug delivery in the treatment of brain tumors. It involves the insertion of BCNU wafers into the resection cavity of patients following surgery [185,186]. These wafers gradually degrade, enabling localized delivery of BCNU to the target area [187]. TMZ was encapsulated within a biologically inert matrix for localized administration to patients with GBM. This encapsulation strategy demonstrated superior efficacy compared to standard therapy alone, resulting in a remarkable increase in overall survival of up to 33 weeks [188]. An injectable enzyme-responsive hydrogel was developed, capable of delivering TMZ and O^6^-BG. This hydrogel demonstrated effectiveness in reducing the recurrence of TMZ-resistant glioma after surgery, while also enhancing the inhibitory efficiency against tumors [189].

Nanoparticle-based delivery offers unique properties that enable targeted delivery to specific cells or tissues [190]. In a recent study, a combination of O^6^-BG formulation with a redox-responsive theranostic superparamagnetic iron oxide nanoparticle (SPION) platform was employed to enhance the intracellular delivery of O^6^-BG to glioblastoma multiforme (GBM) cells while minimizing drug accumulation in healthy tissues [191]. This improved formulation of O^6^-BG showed a significant decrease in MGMT activity and enhanced the cytotoxic effect of TMZ in vitro. Additionally, in an orthotopic primary human GBM xenograft mouse model, it resulted in a three-fold increase in survival compared to untreated controls [191]. Another formulation involved nanoparticles (NPs) coated with a pH-sensitive polymer and a modified analog of MGMT inhibitor, specifically dialdehyde-modified O^6^-benzylguanosine (DABGS). This formulation demonstrated a remarkable inhibition of MGMT activity and enhanced the cytotoxicity of TMZ in vitro [192]. Furthermore, a systematic nano delivery platform (SCL) was developed to encapsulate the p53 gene, facilitating the targeted delivery of p53 to brain tumor tissue. This delivery system successfully depleted MGMT levels and significantly enhanced the therapeutic efficacy of TMZ [193]. The SCL exhibited a notable improvement in the therapeutic effect of TMZ. The CRISPR/Cas9 system, in conjunction with lipid-polymer hybrid nanoparticles (LPHNs-cRGD), was employed to achieve an efficient delivery of pCas9/MGMT plasmids into glioblastoma cells. This delivery approach successfully downregulated MGMT expression levels, resulting in increased chemotherapy sensitivity of tumor cells [194].

In addition, gene delivery can be utilized to efficiently deliver genes to the desired cells. Retroviral and lentiviral vectors expressing inhibitor-resistant MGMT mutants have been utilized to protect against the myelosuppressive toxicity of chemotherapy drugs and prevent therapy-related secondary hematopoietic malignancies [195,196,197,198,199,200].

### 6.4. Targeting the Expression of hMGMT

Several strategies have been developed to target the expression of MGMT. Among them, the identification of microRNAs (miRNAs) that regulate MGMT expression by degrading MGMT mRNA shows promise as an innovative treatment approach to enhance TMZ sensitivity in patients with unmethylated MGMT [38]. Notably, miRNAs such as miR-142-3p, miR-181d, miR-370-3p, and others have been discovered to downregulate MGMT expression and enhance sensitivity to TMZ in GBM cell lines [38,201,202,203,204,205,206,207]. Another promising strategy is the use of siRNA to target MGMT. The combination of TMZ with the MGMT–siRNA/liposome complex has shown a strong synergistic antitumor effect [24,208]. A small-molecule compound, EPIC-0412, was discovered to enhance the chemotherapeutic effect of TMZ by epigenetically silencing the expression of MGMT. It achieved this by targeting two key pathways: the p21-E2F1 DNA damage repair axis and the ATF3-p-p65-MGMT axis [209]. These findings provide compelling evidence for the potential of combining epigenetic drugs to enhance sensitization to TMZ in GBM patients.

The use of oncolytic viruses is indeed a promising strategy to downregulate the expression of MGMT. For example, the overexpression of adenovirus E1A has been shown to efficiently inhibit the promoter activity of MGMT, potentially reducing chemoresistance [210,211].

### 6.5. Others

Autoantibodies against MGMT were detected in patients’ serum [212]. The researchers screened the most responsive peptides using these autoantibodies and discovered that these peptides conferred resistance to TMZ in glioma cells both in vivo and in vitro [213]. This finding suggests that monoclonal antibodies targeting these peptides could serve as a novel strategy to overcome resistance in GBM cases with unmethylated MGMT promoters to alkylating agents. The new agent-KL-50 has been found to induce cell killing selectively in MGMT-silenced tumors, independent of mismatch repair (MMR). It creates a dynamic DNA lesion that can be repaired by MGMT. However, in MGMT-deficient conditions, this lesion slowly evolves into an interstrand cross-link, leading to MMR-independent cell death. Notably, this process exhibits low toxicity in both in vitro and in vivo settings [214]. This agent represents a novel approach in designing chemotherapeutics that exploit specific DNA repair defects. All of the mentioned strategies are summarized in Table 3.

## 7. The Crosstalk of MGMT with Other DNA Repair Pathways

The attack of alkylating agents on DNA can lead to various types of lesions on the heterocyclic bases or backbone. Repairing methylated DNA adducts involves several pathways, primarily the base excision repair (BER) pathway, the family of AlkB homolog proteins (ALKBH), and MGMT (Figure 5) [5,14,30]. The BER pathway plays a critical role in repairing the main N-alkylation DNA adducts, such as N3-methyladenine, N3-methylguanine, and N7-methylguanine, with the first step of this process involving the alkyladenine-DNA glycosylase (AAG, MPG) [5,14,30,215]. Members of the ALKBH family are responsible for the demethylation of N1-methyladenine and N3-methylcytosine. In cases where the removal of m^6^G fails, the resulting m^6^G:T mispair is recognized by the MMR system, leading to a variety of downstream effects [5,14,30]. However, it remains unclear how the different repair pathways collaborate and compete with each other.

### 7.1. MGMT and BER

MGMT and BER are the primary pathways to process DNA lesions induced by alkylating agents. As of now, no glycosylase capable of recognizing m^6^G has been discovered. AAG, on the other hand, specifically recognizes N-alkylated purines and does not compete with MGMT. Nevertheless, there are reports suggesting that MGMT activity is regulated by the BER protein-PARP, which plays a critical role in BER for processing N-alkylpurines [30]. The PARP-mediated PARylation of MGMT, induced by alkylating agents, enhances its binding to chromatin and its ability to facilitate the removal of m^6^G adducts from DNA (Figure 6) [98,99]. This indicates a significant interplay between PARP and MGMT in the repair process.

### 7.2. MGMT and Nucleotide Excision Repair (NER)

The NER pathway plays a crucial role in repairing bulky helix-distorting DNA adducts, like cyclobutene-pyrimidine dimers induced by UV light [216]. Experimental evidence suggests that cells expressing both MGMT and NER proteins efficiently repair O^6^-ethylG, indicating a collaborative effort between MGMT and NER in processing this type of DNA damage [217]. The proposed mechanism involves NER proteins opening up the tightly packed chromatin, thereby aiding MGMT in locating and addressing the DNA adducts [217]. In a recent report, it was discovered that MGMT also collaborates with NER in processing O^6^-carboxymethylG, which has implications in colorectal cancer development and is associated with meat consumption (Figure 6) [218]. Interestingly, the expression of hMGMT in *E. coli* has been observed to hinder the removal of m^4^T by NER, likely due to the significantly slower removal of m^4^T by hMGMT [52]. Notably, alkyltransferase-like proteins (ATLs) that share sequence similarities with the C-terminal domain of MGMT are highly conserved across all three domains of life. When ATLs bind to O^6^-alkylG in DNA, they form a complex that is readily recognized by NER to promote the repair of O^6^-alkylG by NER, effectively obstructing the repair process by MGMT [13,219,220].

### 7.3. MGMT and MMR

The expression of MGMT plays a critical role in preventing cell death induced by alkylating agents, since it directly converts m^6^G back to G. Additionally, MMR also has a significant impact on generating cytotoxic effects triggered by m^6^G lesions [221]. Deficiency in MMR results in resistance against these effects, both in vitro and in vivo [222,223]. Unrepaired m^6^G is highly stable and tends to pair with Thymine (T) instead of G. This m^6^G:T mispair is extremely mutagenic, and can be recognized by the MSH2/MSH6 heterodimer of the MMR pathway [224]. The persistence of m^6^G on the template strand leads to the formation of m^6^G:T mispairs during MMR-directed strand resynthesis, initiating repeated cycles of futile repair [30,221,225,226]. Consequently, unproductive replication cycles across m^6^Gs create unreplicated gaps that interfere with DNA replication in the subsequent S-phase, generating double-strand breaks (DSBs) that ultimately cause cell cycle arrest and cell death (Figure 5) [30,221,225,226]. The futile cycle model finds support through in vitro experiments, which demonstrate that the cytotoxicity of m^6^G takes place during the second cell cycle following treatment [221,225,227]. Circular DNA substrates containing m^6^G/T mismatches, rather than regular G/T mismatches, elicit a MMR-dependent preferential recruitment of ATR-ATRIP, leading to ATR activation (Figure 5) [228]. The observation that the MSH2G674A or MSH6T1217D missense mutants are unable to function in MMR but can still bind to mismatches and initiate apoptosis in response to m^6^G lesions supports the notion that MSH2-MSH6 complexes serve as DNA damage sensors [229,230]. This suggests that the excision of DNA lesions is not necessary for the DNA damage response function of MMR. These findings indicate that ATR activation does not require abortive excision and resynthesis cycles. This direct signaling model proposes that MMR proteins serve as a protein scaffold at m^6^G/T lesions, facilitating the direct recruitment and activation of a global damage response (Figure 5) [221]. The two models of m^6^G-induced cell death are consistent with the following observations: in the absence of MGMT, the MMR pathway is essential for recognizing m^6^G/T mismatches, leading to the induction of DNA strand breaks and ultimately triggering m^6^G-induced cell death. However, the direct signaling model fails to explain why an MMR protein-scaffold at m^6^G/T in the first S-phase does not lead to cell cycle arrest or cell death only after the second S-phase. One possible resolution could entail a combination of these two mechanisms, each contributing to a different outcome [221]. This proposition finds support in data obtained from human pluripotent stem cells (hPSCs). In hPSCs, the apoptosis response induced by an alkylating agent occurs within a few hours, whereas no observable downstream effectors activation is induced by the MMR-signaling complex [221,231,232].

The data obtained from *Xenopus laevis* eggs, where plasmid DNA was exposed to alkylating agents under non-replicating conditions, suggest that concurrent BER and MMR processes on the same DNA molecule may inadvertently lead to the formation of DSBs [233]. This occurrence arises when the repair intermediates of BER and MMR pathways encounter each other, and it may represent an additional mechanism for TMZ-induced cytotoxicity in non-dividing or quiescent cells [233]. This phenomenon is referred to as the “Repair Accident” model [234].

### 7.4. MGMT and DSB Repair

Concomitant with DNA replication, MMR gives rise to DSBs induced by m^6^G lesions, which are responsible for provoking apoptosis signaling [30,39,127]. Considering the crucial role of DSBs in m^6^G induced cytotoxicity, DSB repair is expected to significantly impact m^6^G-induced chromosomal changes. This is supported by the finding that m^6^G lesions caused higher aberration frequencies in cells with ATM deficiency [235]. Thus, MGMT serves as a key defense against clastogenicity by O^6^-methylating agents, acting in concert with MMR, BER, and DSB repair (Figure 5) [30]. After DSBs are formed, they undergo repair through homologous recombination (HR) or non-homologous end joining (NHEJ). BRCA2 plays a crucial role in HR and interacts with hMGMT to facilitate the degradation of methylated or benzylated hMGMT [236].

The alkylation of hMGMT and alkylating damage further enhance the degradation of BRCA2 (Figure 6), indicating a crosstalk between MGMT and HR [236].

## 8. Conclusions

Biochemical studies and crystal structures of MGMT reveal the repair mechanism of hMGMT. Additionally, cell-based assays and animal studies unveil the cancer prevention function of MGMT. Interestingly, when the two domains of hMGMT are expressed separately, N-hMGMT demonstrates Zn^2+^-dependent DNA repair activity in vitro, effectively repairing m^6^G DNA damage but not methylphosphotriester damage [57]. This raises questions about the physiological significance of N-hMGMT and how it repairs m^6^G. Since all the residues responsible for DNA binding and repair are located in hMGMT-C [54,55,58], the mechanism by which m^6^G is repaired by N-hMGMT remains a fascinating aspect to explore. Notably, the active site of hMGMT-C is buried inside the protein [54,55], which prompts the question of how O^6^-BG can be recognized and rapidly located in the active site of hMGMT-C. The fate of alkylated MGMT is still ambiguous. Evidence suggests that alkylated MGMT will be ubiquitinated and degraded [83,84,85], other accumulating evidence shows that alkylated MGMT becomes a transcription regulator to regulate the expression of genes [90,91]. The mechanisms governing the diverse outcomes of alkylated MGMT remain unclear.

MGMT inhibitors, despite promising preclinical results, have not exhibited significant benefits in clinical settings for tumor patients treated with alkylating agents. The main challenge is the increased risk of severe hematological toxicity [13,24,169,175], limiting widespread clinical application. Researchers explore alternative strategies like cancer-selective inhibitors, MGMT expression downregulation, autoantibodies, and localized drug delivery to attenuate MGMT activity and minimize normal cell exposure [13,24,37,39]. While showing promise in cells and animal models, further research and clinical trials are needed to refine these strategies for enhanced efficacy in alkylating agent treatments with minimal harm to healthy tissues.

Alkylating agents exert their effects by inducing various types of DNA lesions, which are repaired by multiple pathways, including the BER pathway, ALKBH, and MGMT [5,14,30]. However, our increasing understanding of the mechanisms of cell killing by alkylating agents reveals that certain cancers, such as gliomas and malignant melanomas, may inherently exhibit resistance to a wide range of these and other anticancer drugs. To address this challenge, an integrated strategy involving MGMT inhibition in combination with cancer chemotherapy is likely necessary. This approach combines the inactivation of MGMT with the inhibition of other repair pathways that protect against the toxicity induced by methylating and chloroethylating agents, along with targeting downstream drug resistance factors [39]. The discovery of crosstalk between MGMT and other DNA repair pathways opens new avenues for improving the clinical response to treatment. For example, the PARylation of MGMT by PARP can regulate the activity of MGMT, potentially enhancing the sensitivity of cancer patients when combining alkylating agents with PARP inhibitors [98,99]. Additionally, the identification of alkylated MGMT and MGMT inhibitors promoting the degradation of BRCA2 presents a promising strategy for targeting homologous recombination proficient cancers with MGMT inactivators in conjunction with DNA crosslinking agents [236]. These novel insights hold significant potential for developing more effective and tailored approaches to cancer treatment, with the aim of overcoming drug resistance and improving therapeutic outcomes for patients. However, further research and clinical studies are essential to validate and refine these strategies before their implementation in clinical practice.

## Figures and Tables

**Figure 1 cancers-16-00331-f001:**
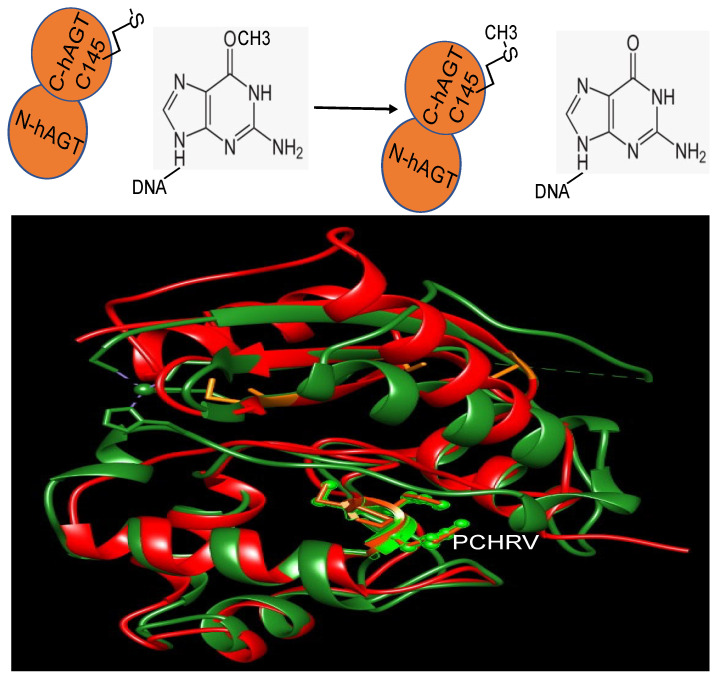
The reaction of MGMT (**top panel**) and the structure overlay of Ada and hMGMT (**bottom panel**). Ada: red; hMGMT: green; the side chains of PCHRV are shown. The overlay is to show the structural similarity of both proteins.

**Figure 2 cancers-16-00331-f002:**
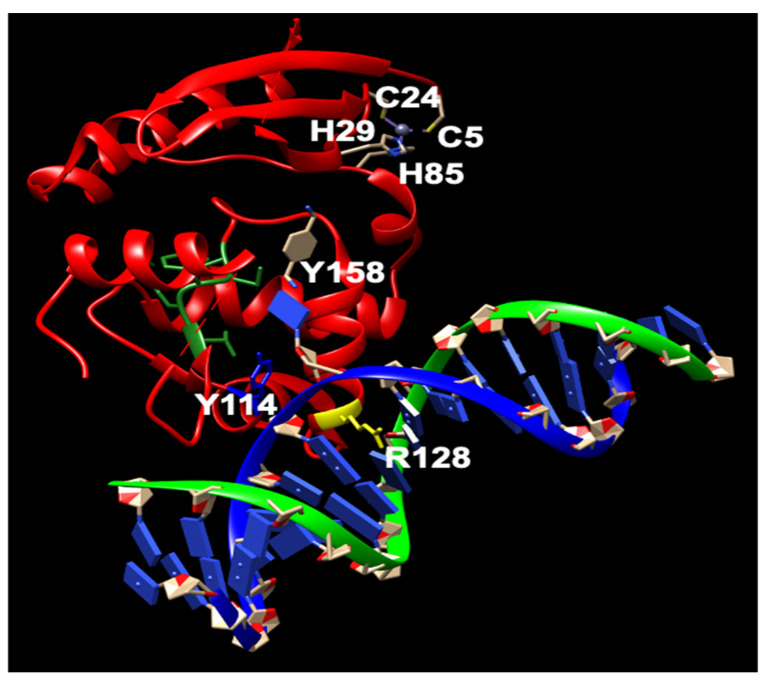
The structure of human MGMT repairing (modified from PDB ID: 1EH7). Y114: blue; R128: yellow; PCHRV: green; hMGMT: red; dsDNA: green and yellow. Nucleotides: blue. The key residues involved in the repair are shown.

**Figure 3 cancers-16-00331-f003:**
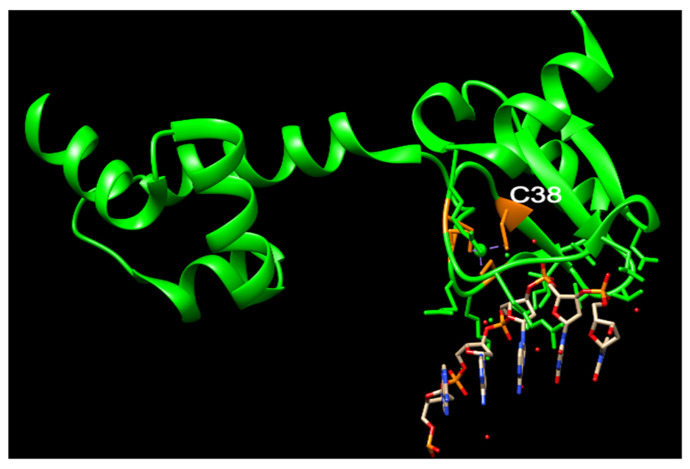
The structure of N-Ada and its interaction with DNA chain (modified from PDB ID: 1U8B). N-Ada: green; zinc cluster with side chains of C38, C42, C69, and C72: orange; phosphodiesterase chains: red and orange. Nucleotides: blue and grey.

**Figure 4 cancers-16-00331-f004:**
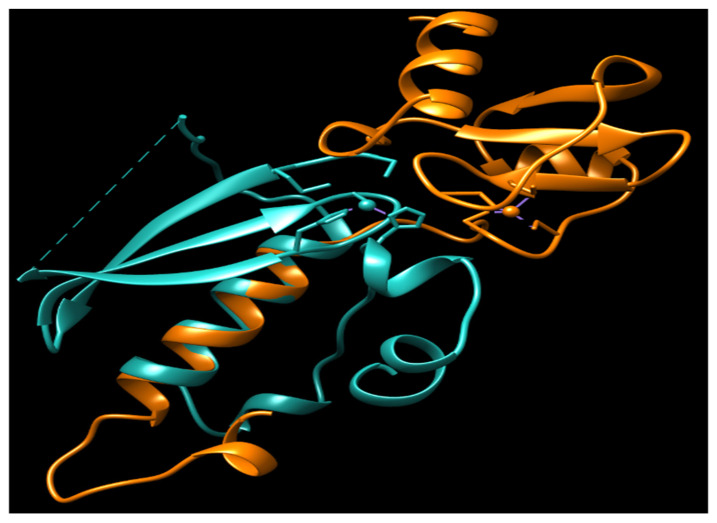
The overlay of N-Ada and N-hMGMT. N-Ada: orange; N-hMGMT: light sea green; Zn^2+^: small ball. Both N-Ada and N-hMGMT contain zinc binding sites and these sites locate in the surface of protein.

**Figure 5 cancers-16-00331-f005:**
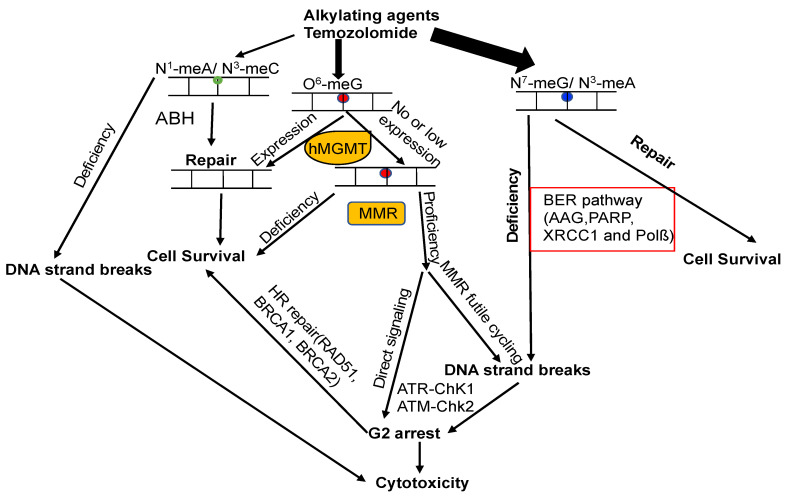
Model of DNA repair pathways involved in DNA alkylation damage. DNA lesions caused by alkylating agents are processed by BER, ALKBH, and MGMT. If m^6^G is unaffected by MGMT during DNA replication, m^6^G forms mismatches with T, generating m^6^G–T pairs. MMR rectifies these mismatches by removing T from m^6^G–T pairs. This futile cycling replication process results in unreplicated gaps opposite the m^6^G lesions, which remain tolerated until the subsequent S-phase (**middle panel**). In this phase, they impede DNA replication, causing DSBs that trigger cell cycle arrest or eventual cell death. Alternatively, mismatch repair proteins recognizing m^6^G/T lesions might function as a scaffold, directly recruiting DNA damage signaling molecules (**middle panel**), leading to the activation of cell cycle checkpoints and/or apoptosis. The circles with different color mean different types of DNA lesions.

**Figure 6 cancers-16-00331-f006:**
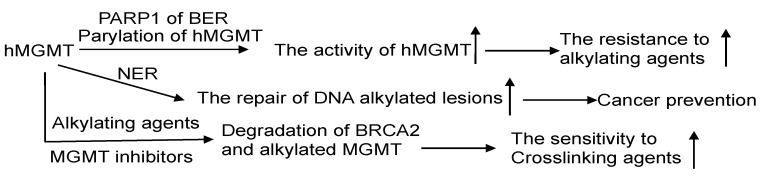
The crosstalk of MGMT with other DNA repair pathways. The interplay of MGMT with other DNA repair mechanisms involves several notable connections. PARP1, a protein in the BER pathway, can perform poly-ADP-ribosylation on hMGMT, resulting in the formation of poly-ADP-ribosylated hMGMT. This modification facilitates the localization of hMGMT within chromatin, enabling efficient repair of DNA lesions. The effectiveness of the NER process can assist MGMT in managing DNA adducts. Through a collaborative effort, MGMT and NER collectively process O^6^-carboxymethylguanine (O^6^-CMG) lesions, thereby thwarting potential carcinogenic outcomes. Intriguingly, the alkylation of hMGMT and the existence of alkylating damage can accelerate the degradation of BRCA2, potentially heightening the sensitivity of cancer cells to the treatment of crosslinking agents. ↑: increase.

**Table 1 cancers-16-00331-t001:** ED50 of MGMT to O^6^-BG.

MGMT	ED50 (µM)
N-hMGMT	0.15 ± 0.01 [57]
N-hMGMT/C-hMGMT	0.20 ± 0.02 [57]
hMGMT	0.24 ± 0.02 [57]
Ogt	600 [73]
Ada	>1 mM (no repair) [73]
YMGMT *	>1 mM (no repair) [73]

* YMGMT: MGMT of Yeast.

**Table 2 cancers-16-00331-t002:** The structure match of N-hMGMT obtained with Phyre2 online server.

#	Template	3D Model	Confidence	Template Information
1	d1qnta2	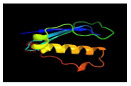	100	Fold: Ribonuclease H-like motif Family: Methylated DNA-protein cysteine methyltransferase domain
2	C1t39A	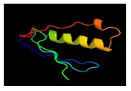	98.7	PDB header: transferase/DNA PDB title: human O^6^-alkylguanine-DNA alkyltransferase covalently crosslinked to DNA
3	C4bhcA	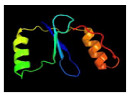	97.7	PDB header: transferase PDB title: crystal structure of the m.tuberculosis O^6^-methylguanine DNA methyltransferase r371 variant

**Table 3 cancers-16-00331-t003:** The list of strategies targeting MGMT.

The Category of Strategies	Compounds or Molecules	Effects and Advantage
Non caner-selective inhibitors	O^6^-BG and O^6^-4-BTG	Efficiently inhibit the activity of MGMT in mammalian cells, animal models and patients [147,148,149,150].
	Meta-substitution of O^6^-BG	Improve the water solubility and more potent inhibitors of MGMT in mammalian cells [168].
	6-(benzyloxy)pyrimidine derivatives	Inhibit the activity of MGMT [13,169].
	Acrolein and chloromethyltriazoles	React with cysteine residues to inhibit the activity of MGMT in cells efficiently [170].
	6-carboxyflfluorescein	Inhibit the activity of MGMT with a non-covalent manner [171].
	Lipoic acid	A natural compound that efficiently inhibits the activity of MGMT and enhances the cytotoxic effects of TMZ in cancer cells that are resistant to TMZ [172].
	Nitric oxide and disulfiram	Inactivate MGMT by interacting with Cys145 [173,174].
Cancer-selective inhibitors	O^6^-BG-Glu and O^6^-BTG-Glu	Conjugate a glucose group with an MGMT inhibitor. They inhibit the activity of MGMT in various cancer cell lines [150,176,177].
	O^4^-benzylfolate	Conjugate a folate group to an MGMT inhibitor. It is a more potent inhibitor than O^6^-BG and enhances BCNU-induced cell death dependent on the expression of the α-folate receptor [179,180].
	O^6^-benzyl-2′-deoxyguanosine	Increase MGMT inhibitory activity in tumor cells and improve the water solubility [179].
	Glucuronic acid linked prodrugs of O^6^-BG and O^6^-benzyl-2′-deoxyguanosine	These prodrugs are stable and less active. O^6^-BG and O^6^-benzyl-2′-deoxyguanosine can be released after the removal of beta-glucuronidase [181,182]. They may be useful in cancer cells that liberate beta-glucuronidase.
Local drug delivery	Gliadel (BCNU wafers)	The first clinical application of polymer drug delivery to treat brain tumors [185,186].
	Encapsulated TMZ with inert matrix	Demonstrated superior efficacy compared to standard therapy alone, resulting in a remarkable increase in overall survival for GBM patients [188].
	Injectable hydrogel capable of delivering TMZ and O^6^-BG	Demonstrated effectiveness in reducing the recurrence of TMZ-resistant glioma after surgery and enhancing the inhibitory efficiency against tumors [189].
	Nanoparticle-based delivery of TMZ and O^6^-BG	These delivery systems successfully deplete MGMT and significantly enhance the therapeutic effect of TMZ in cancer cells and animal models [191,192,193,194].
Targeting the expression of MGMT	miRNAs including miR-142-3p, miR-181d, miR-370-3p	Downregulate the expression of MGMT and enhance the sensitivity to TMZ in GBM cell lines [38,201,202,203,204,205,206,207].
	EPIC-0412	Enhances the chemotherapeutic effect of TMZ by epigenetically silencing the expression of MGMT [209].
	Oncolytic viruses	Downregulate the expression of MGMT [210,211].
Others	Autoantibodies	May overcome the resistance to TMZ in glioma cells [213].
	New agent-KL-50	Induce cell killing selectively in MGMT-silenced tumors, independent of MMR [214].

## Data Availability

The data presented in this study are available in this article.

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
