# Peer review of "The Versatile Attributes of MGMT: Its Repair Mechanism, Crosstalk with Other DNA Repair Pathways, and Its Role in Cancer"

_cancers, 2024, doi:10.3390/cancers16020331_

Round 1

Reviewer 1 Report

Comments and Suggestions for Authors

The review by Fang is a very comprehensive item focusing on the role of MGMT in the scope of potentially damaging events and their pathological consequences. While lots have already been said and written about the clinical potential of MGMT, the look into its pure chemical/structural properties and stepwise explanations towards pathological aspects presented in this review is valuable.

My point for reconsideration:

1. Title. As the text states, there are 3 goals in the ms. The title is around just one of them. It rather should be more precise or general.

2. If those 3 pillars organize the paper, the titles of the main chapters should also reflect that, leaving the more specific subtitles.

3. Lines 40-44. The last sentence is completely unnecessary. First its the linguistic repetition, second, it undermines the value of the review. The first part clearly defines the goals of the review.

4. The pictures in figure 1 and 4 are somehow deformed. Figure 3 is not sharp.

5. Table 2 is not a table but a screenshot. Should be edited, e.g. to proper table removing unnecessary elements

6. Part 5. (line 223-280) should be expanded slightly, and maybe a summarizing table added.

7. The Conclusions section is too long - should be more concise, ev. important information transferred to the text or included in the summarizing table/graph.

8. However I get the idea of the ms, I suppose it may not be as clear directly to the readers. The author should, probably in the introduction, ev. in summary, add 1-2 sentences explaining it.

9. The part about crosstalk with other repair mechanisms contains a comprehensive description of the subject and well-designed figures (could be upgraded graphically). However, I would reconsider its usefulness in the ms or placement in the text. It somehow is a separate subject but, as said earlier, also of value. Maybe some introductory sentences linking better with the previous parts?

Comments on the Quality of English Language

The ms requires some small punctuation/spelling/grammatical/stylistic mistakes. Needs one thorough reading. Especially in the context of some understandable but colloquial expressions, like "chemical insults [...] constantly attack [...] DNA", "the fate of alkylated MGMT".

Some specific mistakes:

line 85 "the structure is shown"

line 493???

Reviewer 2 Report

Comments and Suggestions for Authors

The review by Fang et al. shows that the crosstalk of MGMT with other DNA repair pathways and its role in cancer. This review demonstrated that analysis of the importance of the presented study in the context of the existing knowledge in the field and discussion of the unique aspects presented in the study concerning the repair mechanisms of MGMT and its interaction with other DNA repair pathways. This review comprehensive evaluation of the structure, function, and repair mechanisms of MGMT. Insightful analysis of the interplay between MGMT and other DNA repair pathways. Thorough review of existing literature and the effective integration of past research to support the arguments presented. The considerable potential of the outlined strategies targeting MGMT for cancer therapeutics.

I have some suggestions for improving the text, and several concerns about some arguments rely heavily on past studies that published long time ago, without providing additional insights or fresh perspectives on the topic. Could benefit from questioning and challenging established ideas in the field to present an innovative perspective.

Some elements of the figures and table captions might not be clearly explained or directly relevant to the arguments presented. A clearer connection between visual elements and textual content is needed to aid reader comprehension.

Author Response

Review 2. The review by Fang et al. shows that the crosstalk of MGMT with other DNA repair pathways and its role in cancer. This review demonstrated that analysis of the importance of the presented study in the context of the existing knowledge in the field and discussion of the unique aspects presented in the study concerning the repair mechanisms of MGMT and its interaction with other DNA repair pathways. This review comprehensive evaluation of the structure, function, and repair mechanisms of MGMT. Insightful analysis of the interplay between MGMT and other DNA repair pathways. Thorough review of existing literature and the effective integration of past research to support the arguments presented. The considerable potential of the outlined strategies targeting MGMT for cancer therapeutics.

I have some suggestions for improving the text, and several concerns about some arguments rely heavily on past studies that published long time ago, without providing additional insights or fresh perspectives on the topic. Could benefit from questioning and challenging established ideas in the field to present an innovative perspective.

Response: Thank for your insightful comments. Basic research in the MGMT field indeed advances gradually, given its maturity, with much emphasis on translational and clinical studies. In my discussion, I delved into two innovative topics: 1) exploring the potential novel role of the N-terminal domain of MGMT, and 2) investigating the interactions between MGMT and other DNA repair pathways to assess if these findings could offer new strategies for enhancing current chemotherapy approaches. As for the uncertain fate of alkylated MGMT, I propose two potential mechanisms and research directions that could aid in reconciling contradictory findings.

Some elements of the figures and table captions might not be clearly explained or directly relevant to the arguments presented. A clearer connection between visual elements and textual content is needed to aid reader comprehension.

Response: Thank for the comments. I revised some sentences and table captions to establish a clearer connection.